# Molecular Confirmation of Raptors from Spain as Definitive Hosts of Numerous *Sarcocystis* Species

**DOI:** 10.3390/ani15050646

**Published:** 2025-02-23

**Authors:** Evelina Juozaitytė-Ngugu, Saulius Švažas, Antonio Bea, Donatas Šneideris, Diego Villanúa, Dalius Butkauskas, Petras Prakas

**Affiliations:** 1Nature Research Centre, Akademijos Str. 2, LT-08412 Vilnius, Lithuania; evelina.ngugu@gamtc.lt (E.J.-N.); saulius.svazas@gamtc.lt (S.Š.); donatas.sneideris@gamtc.lt (D.Š.); dalius.butkauskas@gamtc.lt (D.B.); 2Ekos Estudios Ambientales S.L.U., Donostia Etorbidea 2, Bajo 2, 20160 Lasarte, Spain; a.bea@ekos-sl.com; 3Navarra Environmental Management GAN-NIK, Calle Padre Adoain 219, 31015 Pamplona, Spain; diegovillanua@yahoo.es

**Keywords:** *Sarcocystis*, raptors, DNA analysis, life cycle, *ITS1*, *28S* rRNA, phylogeny

## Abstract

*Sarcocystis* species are globally distributed apicomplexan parasites of reptiles, birds, and mammals with a complex life cycle requiring two hosts. Sarcocysts are formed mainly in the muscles of intermediate hosts (prey), while oocysts and sporocysts are found in the intestines of definitive hosts (predator and scavenger). Birds are still insufficiently investigated as definitive hosts of *Sarcocystis* parasites. This is the first study to molecularly identify the natural definitive hosts of *Sarcocystis* species, examining a wide range of raptor species with different feeding ecology and behaviour. Overall, 40 birds belonging to the Accipitridae, Falconidae, and Strigidae families from Spain were analysed. *Sarcocystis* species were identified using Sanger sequencing. Twelve known *Sarcocystis* species and three genetically new organisms were confirmed in the intestines of raptors. Identified *Sarcocystis* species use birds, rodents, or predatory mammals as intermediate hosts. Among *Sarcocystis* species uncovered, *S*. *glareoli* form sarcocysts in the brain of rodents, and *S*. *halieti* is potentially pathogenic for birds. The detection of certain *Sarcocystis* species was consistent with ecological data on the diet of raptors analysed.

## 1. Introduction

Birds from Accipitridae, Falconidae, and Strigidae families are at the top of the food chain and play a key role in their ecosystems [1,2,3]. Most raptors are flagship and umbrella species with very different behaviours, feeding ecology, habitat selection, and migratory patterns [4]. In Spain, 26 species of diurnal raptors and 8 nocturnal ones were recorded [5]. For instance, Spain is home to the largest European population of Eurasian Griffon Vultures (*Gyps fulvus*) with an estimated 30,100–36,500 individuals [6]. When raptor populations decline, the number of prey species increases, creating an imbalance in the ecosystem. Research on avian pathogens in raptors is important for several reasons, it can provide useful data for monitoring ecosystem health, for assessing the health status of raptor populations, and for the role of these birds in the dissemination of some important pathogens [5,7,8]. Endoparasites found in raptors include protozoans, roundworms (nematodes), spiny-headed worms (acanthocephala), flukes (digenetic trematodes), tapeworms (cestodes), and tongue worms (pentastomida). For instance, in Germany, 31.4% of 194 raptors examined were infected with coccidia in mucosal smear samples [9].

*Sarcocystis* spp. are globally distributed apicomplexan parasites with an obligatory two-host life cycle [10]. These parasites can infect a variety of reptiles, birds, and mammals, including humans [11]. Sarcocysts are found mainly in the muscles or CNS of intermediate hosts (IHs), while the endogenous sporulation of oocysts takes place in the intestine of the DHs [12]. *Sarcocystis* species are morphologically characterised and described in IHs, while sporocysts of parasite species found in DHs can be differentiated only by molecular tools [13,14,15]. Some *Sarcocystis* species are zoonotic, and others cause economic losses for farmers or are pathogenic for wildlife animals [10,11,12].

The genus *Sarcocystis* encompasses around 220 species, and for approximately 10% of them, raptors serve as their definitive hosts (DHs) [12,16,17,18]. Birds from Accipitridae, Falconidae, and Strigidae families may spread more than 20 valid *Sarcocystis* spp. that infect birds of different orders, small mammals (rodents, lagomorphs, etc.), and carnivores [12]. However, there is still a lack of studies on the role of birds in the transmission of various *Sarcocystis* spp. Previously, DHs were enclosed by performing laboratory transmission experiments [19,20,21,22]. To date, due to stricter ethical requirements when using raptors as experimental hosts, potential DHs are increasingly being identified using DNA analysis techniques [13,18,23,24,25,26,27]. *Sarcocystis* spp. in the DH usually cause no symptoms, but their presence might lead to weight loss, diarrhoea, or nausea [11,12].

The aim of the current study was to identify *Sarcocystis* spp. in intestinal samples from raptorial species collected in Spain by molecular methods.

## 2. Materials and Methods

### 2.1. Collection of Raptor Samples

A total of forty raptors, two Eurasian Goshawks (*Accipiter gentilis*), two Eurasian Sparrowhawks (*Accipiter nisus*), one Golden Eagle (*Aquila chrysaetos*), eleven Common Buzzards, four Western Marsh Harriers (*Circus aeruginosus*), two Eurasian Griffon Vultures, five Black Kites (*Milvus migrans*), three Red Kites (*Milvus milvus*), one Lesser Kestrel (*Falco naumanni*), seven Common Kestrels (*Falco tinnunculus*), and two Eurasian Eagle-Owls (*Bubo bubo*) were collected in 2023. Most of these birds, i.e., thirty, belong to the Accipitridae family, while eight and two birds are representatives of Falconidae and Strigidae, respectively. The birds were admitted to the Wildlife Recovery Centre in Ilundain (Navarra) (Spain). The samples were taken by the centre’s veterinary staff while carrying out their routine diagnostic protocol for the cause of death of the birds, which were either brought to the centre as dead specimens or died there. This centre is under the jurisdiction of the Government of Navarra and is managed by a public company, GAN-NIK. The permission to collect wild birds was granted by the Government of Navarra in accordance with Resolution Nr. 278/2022, 20 December 2022. Raptors were kept frozen at −20 °C until the birds were dissected.

### 2.2. Preparation of Intestines of Sarcocystis spp.

The isolation of *Sarcocystis* oocysts/sporocysts from the intestine of DHs was performed at the Anatomy, Embryology, and Animal Genetics Laboratory of the University of Zaragoza, in Spain. *Sarcocystis* spp. were extracted from the entire intestine of each raptor using a modified protocol [28]. However, the laboratory in which the preparation of bird intestines was carried out lacked some of the necessary facilities; therefore, it was not possible to implement all the recommendations proposed by Verma et al. [28]. The detailed procedure for isolating oocysts/sporocysts of *Sarcocystis* spp. is described below.

The entire intestine was removed from the bird and cut lengthwise. The intestine was placed in a laboratory container, suspended in 120 mL of distilled water (dH_2_O), and crushed using a hand mixer at top speed for 4–5 min. The homogenate was centrifuged for 6 min at 1000 rpm in two 15 mL centrifuge tubes. The supernatant was discarded and the remaining homogenate from the laboratory container was added to 15 mL tubes; this process was repeated three to five times. The remaining precipitate was suspended in HBSS (Hanks’ Balanced Salt Solution), filtered through two layers of cheesecloth, and emulsified in a 5.25% sodium hypochlorite (bleach) solution (1:1 ratio) in a cold bath for 30 min. The homogenate was centrifuged at 1000 rpm for 6 min and the obtained supernatant was discarded. Sediments were suspended in dH_2_O and centrifuged in the same conditions; this process was repeated until the smell of bleach (chlorine) was gone. The isolation of DNA was performed on all bird samples. A total of 400 µL of re-suspended sediments was taken from each sample and used for DNA extraction.

### 2.3. Molecular Analysis of Sarcocystis spp.

Total DNA was extracted from samples using the GeneJET Genomic DNA Purification Kit (Thermo Fisher Scientific Baltics, Vilnius, Lithuania) according to the manufacturer’s instructions. Samples were kept frozen at −20 °C until further analysis.

Nested PCR was performed using a wide range of primers (Table 1) targeting internal transcribed sequences (*ITS1*) and large ribosomal subunit (*28S* rRNA) sequences. PCR reactions were carried out using DreamTaq PCR Master Mix (Thermo Fisher Scientific Baltics) according to the manufacturer’s instructions. The following cycling conditions were used: 5 min at 95 °C, 35 cycles of 35 s at 94 °C, 45 s at 55–65 °C depending on the primer pair (Table 1), 60 s at 72 °C, and 5 min at 72 °C. Positive (DNA of *Sarcocystis* species tested) and negative (nuclease-free water instead of DNA template) controls were used in each PCR. Amplified products were observed in 1% agarose gel and then purified before sequencing with Exonuclease I and FastAP Thermosensitive Alkaline Phosphatase (Thermo Fisher Scientific Baltics). Products were sequenced directly with the 3500 Genetic Analyzer (Applied Biosystems, Foster City, CA, USA) using the same forward and reverse primers as for PCR. The obtained *ITS1* and *28S* rRNA sequences were deposited in GenBank with accession numbers PQ932237–PQ932324 and PQ932346–PQ932402, respectively.

### 2.4. Phylogenetic Analysis

Phylogenetic analyses were performed by inferring phylogenetic relationships of the *ITS1* and *28S* rRNA sequences obtained. Multiple sequence alignments were performed using the MUSCLE algorithm incorporated into MEGA11 v. 11.0.13 software [32]. Gaps/missing data were treated using the partial deletion option. The selection of the nucleotide substitution model that best fitted the analysed data was achieved with TOPALi v2.5 [33], according to the generated lowest scores of the Bayesian Information Criterion. TOPALi v2.5 was used to reconstruct phylogenetic relationships under Bayesian inference. Bayesian analyses were carried out in two runs, using one million replicates at a sample rate of 10 and 25% burn-in values.

## 3. Results

### 3.1. Molecular Identification of Sarcocystis Species

In the present study, we screened for some molecularly well-characterised *Sarcocystis* spp. using predatory mammals, birds, and rodents as their DH. Of all the *Sarcocystis* spp. forming sarcocysts in the muscles of predatory mammals, only *S*. *arctica* was identified in two samples of intestinal scrapings from raptors (Table 2). The *ITS1* sequences of *S*. *arctica* shared 96.4–100% similarity with other isolates of this species and 88.2–89.4% similarity to those of *S*. *felis* (AY190081-2, KC160214) from domestic cats and Geoffroy’s cat (*Leopardus geoffroyi*).

Based on species-specific primers targeting *ITS1*, five avian *Sarcocystis* species, *S*. *columbae*, *S*. *cornixi*, *S*. *halieti*, *S*. *kutkienae*, and *S*. *turdusi*, were established (Table 2). Furthermore, we obtained *28S* rRNA sequences of *Sarcocystis* sp. ex *Corvus corax* and *Sarcocystis* sp. 22AvEs1 which, by phylogenetic results, seems to employ birds as their IHs. Our five 616 bp sequences amplified with the help of SgraupaukF and SgraupaukR primers demonstrated 100% identity to *Sarcocystis* sp. ex *Corvus corax* (MZ701977) from the Common Raven (*Corvus corax*) and 99.4% similarity with *S*. *corvusi* (JN256118) from the Western Jackdaw (*Corvus monedula*). After examining the same bird sample and using Ssprod2F/Ssprod2R and SgraupaukF/SgraupaukR primer pairs, we obtained 622 bp and 616 bp *28S* rRNA sequences, respectively. The following sequences were 100% identical to each other and showed the highest (98.8–98.9%) similarity with that of *S*. *turdusi* (JF975682) from Common Blackbird (*Turdus merula*).

Finally, *28S* rRNA and *ITS1* sequences of four genetically different *Sarcocystis* species were detected, i.e., *S*. *glareoli*, *S*. cf. *strixi*, *Sarcocystis* sp. Rod6, and *Sarcocystis* sp. Rod7 showed the closest similarity to those of *Sarcocystis* spp. with rodents as intermediate hosts (*S*. *funereus*, *S*. *glareoli*, *S*. *jamaicensis*, *Sarcocystis* sp. Rod3, *S*. cf. *strixi*, *S*. *strixi*) (Table 2). The *ITS1* sequences of *S*. *glareoli* obtained in this work shared 99.8–100% of their identity to those of the same species and 98.3–98.5% similarity with those of *Sarcocystis* sp. Rod3. Twelve *ITS1* sequences amplified with the GsSglajamF1/GsSglajamR1 primer pair were not designated as *S*. *glareoli*, as genetic similarity with this species in the region analysed was 95.8–96.5%. Two of the following 511 bp sequences attributed to *Sarcocystis* sp. Rod7 were 100% identical to each other and demonstrated 95.8–96.3% to 10 other sequences assigned to *Sarcocystis* sp. Rod6. The differences between the four genotypes of *Sarcocystis* sp. Rod6 were 0.2–0.8%. By using the SgraupaukF/SgraupaukR primer pair, the obtained *28S* rRNA sequences of *S*. *glareoli*, *Sarcocystis* sp. Rod6, and *Sarcocystis* sp. Rod7 shared very high (99.4–99.7%) similarity with those of *S*. *jamaicensis* and *Sarcocystis* sp. Rod3. Lastly, using GsSstrF1/GsSstR1 primers targeting *28S* rRNA, we obtained 20 sequences displaying 99.2 –100% similarity between each other, 99.6–100% similarity with that of *S*. cf. *strixi* (OQ557459) from the yellow-necked mouse (*Apodemus flavicollis*), 99.1–99.4% similarity with that of *S*. *strixi* (MF162316) from experimental host laboratory mice, and 97.4–98.0% similarity with those of *S*. *funereus* (MW349707, OR725602, OR726006, PP350819) from the blood of bank voles (*Clethrionomys glareolus*) and intestinal scrapings of Boreal Owls (*Aegolius funereus*). Based on the highest genetic similarity values, these 20 sequences were assigned to *S*. cf. *strixi*.

Three and six *28S* rRNA sequences amplified with Ssprod2F/Ssprod2R and SgraupaukF/SgraupaukR primer pairs, respectively, had double or poly peaks. Based on BLAST (http://blast.ncbi.nlm.nih.gov/, accessed on 5 December 2024) analysis, the mentioned sequences showed the highest similarity with *Sarcocystis* spp.; however, as the sequences were not pure, they were not used in further analyses.

### 3.2. Phylogenetic Placement of Identified Sarcocystis Species

Phylogenetic trees were constructed based on the used primer pairs. The phylogenetic relationships of *Sarcocystis* species with birds as IHs, which were amplified using species-specific primers targeting *ITS1*, are shown in Figure 1a. Five identified species, *S*. *columbae*, *S*. *cornixi*, *S*. *halieti*, *S*. *kutkienae*, and *S*. *turdusi*, with high support, were grouped with other isolates of the same species. Using primers specific for *S*. *arctica*, forming sarcocysts in the muscles of predatory canids, the two obtained *ITS1* genotypes of *S*. *arctica* were placed together with other genotypes of this species and were most closely related to *S*. *felis* (Figure 1b). Based on *ITS1*, *Sarcocystis* sp. Rod6 and *Sarcocystis* sp. Rod7 formed a sister clade consisting of *S*. *glareoli* and *Sarcocystis* sp. Rod3 isolates (Figure 1c). In the obtained phylogenetic tree, there was no discrimination between *S*. *glareoli* and *Sarcocystis* sp. Rod3. Meanwhile, maximum support was given to separate the four obtained genotypes of *Sarcocystis* sp. Rod6 from *Sarcocystis* sp. Rod7 (Figure 1c). When primers were used to specifically amplify organisms similar to *S*. *strixi*, the four different obtained sequences of *S*. cf. *strixi* were placed into one cluster together with *S*. cf. *strixi* from the yellow-necked mouse (*Apodemus flavicollis*) and were differentiated from *S*. *strixi* from the experimental host interferon gamma gene knockout (KO) mouse (Figure 1d). Based on *Sarcocystis* genus-specific primers, our sequences of *S*. cf. *strixi* were also more closely related to *S*. cf. *strixi* than to *S*. *strixi* (Figure 1e,f). Using relatively short *28S* rRNA sequences, it was not possible to distinguish the identified *S*. *glareoli*, *S*. sp. Rod6, and *S*. sp. Rod7 from each other and from other closely related organisms (Figure 1e). The obtained *28S* rRNA sequences of *S*. sp. ex *Corvus corax* were grouped together with that of an unnamed *Sarcocystis* organism originally detected in the muscles of the Common Raven and formed a sister cluster to *S*. *halieti* (Figure 1e). Our *28S* rRNA sequences of *S*. sp. 22AvEs1 clustered with those of two avian *Sarcocystis* species, *S*. *fulicae* and *S*. *turdusi* (Figure 1f).

### 3.3. Distribution of Sarcocystis spp. in Raptors Analysed

By means of molecular methods, *Sarcocystis* spp. were confirmed in 33 of 40 (82.5%) samples investigated. The prevalence of *Sarcocystis* spp. in certain host species varied from 50% to 100%; however, in most cases, only several individuals of raptorial species were studied (Table 3). The detection rates of *Sarcocystis* spp. were 86.7% (26/30), 62.5% (5/8), and 100% (2/2) in birds of Accipitridae, Falconidae, and Strigidae families, respectively.

The number of identified *Sarcocystis* species per sample varied from one to six, which was established in a single Black Kite (Appendix A). More than one *Sarcocystis* species was identified in 72.5% of birds. The average number of *Sarcocystis* species detected per sample ranged from one for Common Kestrel to five for Eurasian Goshawk (Table 3). Overall, the average number of species established was 1.9, while in Accipitridae, Falconidae, and Strigidae, it accounted for 3.2, 1.8, and 2.0, respectively.

All 12 *Sarcocystis* species identified in the current study were detected in examined Accipitridae; *S*. *columbae*, *S*. *cornixi*, *S*. *glareoli*, *S*. *halieti*, *S*. cf. *strixi*, and *S*. sp. Rod6 were identified in two species of Falconidae; and *S*. *columbae*, *S*. *cornixi*, and *S*. cf. *strixi* were confirmed in two Eurasian Eagle-Owl individuals representing the Strigidae family. The largest number of different *Sarcocystis* species, as many as nine, was found in Common Buzzard. Afterwards, seven parasite species each were identified in Eurasian Goshawks and in Red Kites.

A similar number of *Sarcocystis* species using birds or rodents as their IHs were identified in birds of the orders Falconiformes and Strigiformes (Figure 2). In general, *Sarcocystis* spp. with birds as IHs compared to those with rodents as IHs were more commonly detected in the small intestines of Accipitridae. The highest proportion of *Sarcocystis* spp. from birds versus parasite species from rodents was observed in the examined *Accipiter* hawks. Specifically, only *S*. *glareoli* forming sarcocysts in the brains of rodents was found in intestinal scrapings of Eurasian Sparrowhawk and Northern Goshawk. By contrast, four and six different avian *Sarcocystis* spp. were established in two *Accipiter* species.

## 4. Discussion

### 4.1. Molecular Identification of Sarcocystis spp. from Raptors in Spain

Generally, the morphometric sizes of a parasite’s sexual stages cannot help to identify the *Sarcocystis* species in the samples tested [10,12]. Recently, advanced molecular methods using PCR have become essential to genetically identify these protists found in the intestines and faeces of the potential DHs [13,18,23,24,25,26,27,34,35]. In most cases, such studies analyse one or two bird species [13,15,18,25,30,36]. There are only a few studies that test a wider range of DH species for *Sarcocystis* spp. In 2022, Rogers et al. [14] examined eight bird species from California, USA, as potential DHs for *Sarcocystis* spp. Furthermore, intestinal scrapings of six species of the family Corvidae have been analysed for *Sarcocystis* spp. in Lithuania [23]. However, the possible determination of pseudoparasitism for some *Sarcocystis* species found in intestinal scrapings of raptors cannot be excluded. Based on previous investigations and data regarding DHs and IHs for some species, the detection of *Sarcocystis* DNA in intestinal scrapings does not necessarily mean that the oocysts/sporocysts were formed in the examined birds. The recovered DNA may also belong to *Sarcocystis* spp. that were in the bird’s feeding carcass [13,23]. Further transmission experiments are necessary to confirm whether the identified *Sarcocystis* species are transmitted by these raptorial birds. Here, we present the first molecularly based identification of *Sarcocystis* spp. in the small intestines of 11 raptorial species in Spain. Their feeding ecology is very different. Eurasian Griffon Vultures are scavengers, while the Eurasian Sparrowhawk and the Northern Goshawk are primarily ornitophagous. Other studied species of raptors are generalist carnivores [1,2,3].

Based on the comparison of the obtained *28S* rRNA and *ITS1* sequences, 12 *Sarcocystis* species (*S*. *arctica*, *S*. *columbae*, *S*. *cornixi*, *S*. *glareoli*, *S*. *halieti*, *S*. *kutkienae*, *S*. cf. *strixi*, *S*. *turdusi*, and *Sarcocystis* sp. ex *Corvus corax*, and potentially new *Sarcocystis* sp. Rod6, *Sarcocystis* sp. Rod7, and *Sarcocystis* sp. 22AvEs1) were identified in the intestinal samples of raptors in Spain (Table 2 and Table 3). Phylogenetic analyses showed that *S*. *arctica*, *S*. *columbae*, *S*. *cornixi*, *S*. *glareoli*, *S*. *halieti*, *S*. *kutkienae*, *S*. cf. *strixi*, *S*. *turdusi*, and *Sarcocystis* sp. ex *Corvus corax* were significantly related to other previously detected isolates of certain species (Figure 1). Furthermore, in the present work, newly identified *Sarcocystis* sp. Rod6, *Sarcocystis* sp. Rod7, and *Sarcocystis* sp. 22AvEs1 were genetically distinguished from other *Sarcocystis* spp. (Table 2, Figure 1). The IHs and DHs of previously known *Sarcocystis* species are listed in Appendix B. According to current knowledge, we determined the new potential DHs for *S*. *arctica*, *S*. *columbae*, *S*. *cornixi*, *S*. *glareoli*, *S*. *halieti*, *S*. *kutkienae*, and *S*. cf. *strixi*.

### 4.2. Identification of Sarcocystis Parasites Forming Sarcocysts in Predatory Mammals

*Sarcocystis arctica* and *S. lutrae* are among the most well-studied *Sarcocystis* species forming sarcocysts in the muscles of predatory mammals [12,23,34,37,38,39,40,41,42,43,44]. Sarcocysts of *S*. *arctica* have been identified in various muscles of the Arctic fox (*Vulpes lagopus*) [37], the domestic dog (*Canis familiaris*) [38,42], the grey wolf (*Canis lupus*) [39,44], and the red fox (*Vulpes vulpes*) [40]. On the basis of molecular data, the White-tailed Eagle (*Haliaeetus albicilla*) [15], the Common Raven (*Corvus corax*) [23], and the Hooded Crow (*Corvus cornix*) [23] have been suggested as potential DHs for *S*. *arctica*. Meanwhile, *S*. *lutrae* is less IH-specific and has been found in the muscles of various mustelids [41], canids [40,43], and procyonids [43]. The DNA of *S*. *lutrae* has been identified in the small intestines of the White-tailed Eagle from the Czech Republic [36] and of the Hooded Crow from Lithuania [23]. In the present study, the DNA of *S*. *arctica* was found in the intestinal scrapings of the Common Buzzard and the Red Kite (Table 3). Hence, raptors and scavengers may be potential DHs of *S*. *arctica*. Meanwhile, we did not detect *S*. *lutrae* in the intestinal scrapings of the raptors analysed. Interestingly, this *Sarcocystis* species was found in the muscles of red foxes from the Baltic States but not from Spain [40]. Further detailed studies on the muscles of carnivorous mammals are required to reveal whether *S*. *lutrae* is prevalent in Spain.

### 4.3. The Role of Raptors in the Transmission of Sarcocystis spp. Using Birds as Their IH

Of the *Sarcocystis* species detected in the present work, six are found in the muscles of various birds [12,13,17,45,46,47,48,49,50,51,52,53]. For the first time, *S*. *columbae* DNA has been detected in the intestinal scrapings of seven new DH species (Table 2 and Table 3). Thus, even 13 bird species may contribute to the transmission of *S*. *columbae* [14,23,25]. In addition, based on the comparison of the obtained *ITS1* sequences, for the first time, *S*. *cornixi* has been identified in the intestinal samples of five new DH species. Currently, this species can employ even 13 bird species as potential DHs (Appendix B) [16,23,25,30]. For the first time, *S*. *halieti* was detected in four new DH species. Hence, even 15 bird species may transmit the pathogenic *S*. *halieti* [13,14,16,23,25,30]. In addition, for the first time, *S*. *kutkienae* has been detected in the intestinal scrapings of the Black Kite (Table 2 and Table 3). Thus, the DNA of this *Sarcocystis* species has been found in the intestinal scrapings of nine bird species (Appendix B) [23,25,30].

Furthermore, this is the first confirmation of potentially new *Sarcocystis* sp. ex *Corvus corax* in the Common Buzzard, the Eurasian Goshawk, and the Eurasian Sparrowhawk. *Sarcocystis* sp. ex *Corvus corax* was originally found in the leg muscles of a single Common Raven from Lithuania and characterised using light microscopy and *18S* rRNA, *28S* rRNA, and *ITS1* sequence analysis [46]. Sarcocysts of *Sarcocystis* sp. ex *Corvus corax* are indistinguishable under a light microscope from those of *S*. *corvusi* and *S*. *halieti* which were also identified in the muscles of birds of the family Corvidae [52]. Hence, electron microscopy analysis of sarcocysts of *Sarcocystis* sp. ex *Corvus corax* is needed for the description of a new *Sarcocystis* species. Based on the comparison of the obtained *ITS1* sequences, a potentially new *Sarcocystis* sp., *Sarcocystis* sp. 22AvEs1, was identified in the intestinal samples of the Western Marsh Harriers in Spain (Table 2). In the phylogenetic trees, *Sarcocystis* sp. 22AvEs1 was most closely related to *S*. *fulicae* and *S*. *turdusi* (Figure 1e,f), forming sarcocysts in terrestrial and water birds [47,54].

In this work, certain *Sarcocystis* species (*S*. *columbae*, *S*. *cornixi*, and *S*. *halieti*) were identified in several bird species belonging to different orders (Table 3). Hence, our study supports earlier observations that avian *Sarcocystis* species have low specificity for their DHs [13,14,23,25,30]. Among the studied raptor species, the Eurasian Sparrowhawk and the Northern Goshawk are primarily ornitophagous. For instance, the Eurasian Sparrowhawk primarily preys on smaller woodland birds, such as Great Tits (*Parus major*), House Sparrows (*Passer domesticus*), tits (family Paridae), finches (family Fringillidae), buntings (genus Emberiza), thrushes (family Turdidae), and starlings (family Sturnidae) [55]. These birds can hunt more than 120 bird species, as well as small mammals [56]. Furthermore, the Black Kite generally snatches small live prey, birds, fish, bats, and rodents [57]. Sometimes, these raptors also feed on waterfowl, especially in the breeding season to feed their young [55]. The Common Kestrel exhibits great trophic plasticity between populations. In Northern Europe, this raptor is a micro-mammal specialist, while in Mediterranean areas, it is a generalist and feeds on passerines, small mammals, insects, reptiles, and amphibians [58,59]. In Spain, the diet of the Common Kestrel consists of insects (found in 89.9% of all samples), birds (7.5% of samples), and mammals (2.5% of samples), though mammals represent the main part of the biomass-fed species comprising 62.3% of the diet [59]. In northwestern Spain, small mammals and reptiles are the most frequent prey in the diet of the Common Buzzard, while amphibians and birds are much less frequent [60]. In this study, we detected *S*. *columbae* and *S*. *cornixi* in the sole representative of the investigated order Strigiformes, the Eurasian Eagle-Owl. In Central Norway, it was shown that birds compose 21.2% of the diet of the Eurasian Eagle-Owl, including bird species that have been proven to be IHs of *S*. *columbae* [61,62]. In Spain, its diet mainly consists of rabbits (*Oryctolagus cuniculus*) [63]. We have identified *S. columbae*, *S. glareoli*, *S. halieti*, and *S.* cf. *strixi* in the Eurasian Griffon Vulture. In Spain, this species feeds almost exclusively on livestock carcasses [64]. In general, the detection and distribution of certain *Sarcocystis* species in raptors in Spain are consistent with their diet preferences.

The DNA of *S*. *calchasi*, *S*. *corvusi*, *S*. *fulicae*, *S*. *lari*, and *S*. *wobeseri* was not detected in the intestinal scrapings of raptors in this study. Three of these species, *S*. *corvusi*, *S*. *fulicae*, and *S*. *lari*, up until now, have only been recorded in Lithuania [49,54,65,66]. Furthermore, DHs of *S*. *corvusi* and *S*. *fulicae* have yet to be disclosed. *Sarcocystis wobeseri* was detected in the muscles of anseriforms [67], larids [49,53], and raptors [68] from Lithuania and the United Kingdom. The DNA of this species has been identified in the intestinal scrapings of Accipitridae [25] and Corvidae [23] birds. The distribution of *S*. *wobeseri* in other countries is still unknown. Previously, pathogenic *S*. *calchasi* has been identified in birds of the orders Columbiformes [69,70,71,72,73,74,75,76,77], Galliformes [78], Piciformes [79], Psittaciformes [70,80,81], and Suliformes [82]. *Accipiter* hawks serve as DHs for this species [14,25,46,69,72]; however, in the current study, only four birds of Eurasian Goshawks and Eurasian Sparrowhawks were investigated. Although pathogenic *S*. *calchasi* has been identified in Europe [25,46,69,70,71,72,73,74,75,76,77], it has not yet been recorded in Spain.

### 4.4. The Role of Raptors in the Transmission of Sarcocystis spp. Using Rodents as Their IHs

In the intestines of raptors, we found that *S*. *glareoli*, *S*. cf. *strixi*, *Sarcocystis* sp. Rod6, and *Sarcocystis* sp. Rod7 use rodents as their IHs (Table 2). Previous investigations have shown that IHs of *S*. *glareoli* and *S*. cf. *strixi* are rodents (Appendix B) [83,84,85], while based on *ITS1* sequences, two potentially new species, *Sarcocystis* sp. Rod6 and *Sarcocystis* sp. Rod7, were most closely related to *S*. *glareoli* (Figure 1). Therefore, we assume that both potentially new species detected can form sarcocysts in the muscles of rodent species, which should be revealed in further studies.

For the first time, *S*. *glareoli* was established in the intestinal samples of five new raptor species. Furthermore, *S*. cf. *strixi* DNA was found in the intestinal samples of nine new DHs for the first time. The previous findings on the IHs and DHs of these two *Sarcocystis* spp. are summarised in Appendix B. In the present study, based on the comparison of the obtained *ITS1* sequences, the DNA of the potentially new *Sarcocystis* sp. Rod6 was detected in the intestinal samples of four raptor species. Meanwhile, *Sarcocystis* sp. Rod7 was identified in intestinal samples of Common Buzzards only (Table 2). It should be highlighted that closely related *Sarcocystis* sp. Rod6 was not identified in the small intestines of the 11 analysed Common Buzzards. We assume that either *Sarcocystis* sp. Rod7 is specialised to be transmitted by the Common Buzzard, or this species is relatively rare in the area under investigation. Based on *ITS1* sequences obtained using the GsSglajamF1/GsSglajamR1 primer pair, of the 11 Common Buzzards examined, *S*. *glareoli* was identified in six birds, whereas *Sarcocystis* sp. Rod7 was identified in two birds. Using the same primer pair (GsSglajamF1/GsSglajamR1), 30 Common Buzzards from Lithuania were tested for *Sarcocystis* spp., and *S*. *glareoli* was detected in 16 birds, while *Sarcocystis* sp. Rod3 was detected in two birds [30]. Furthermore, *Sarcocystis* sp. Rod3 was not found in other raptorial birds from Spain. So, *Sarcocystis* sp. Rod3 might use IHs that are not present in Spain; likewise, *Sarcocystis* sp. Rod7 might employ IHs that are not prevalent in Lithuania.

In previous investigations, very low (less than 0.5%) genetic differences have been demonstrated comparing partial *28S* rRNA sequences of *S*. *glareoli*, *S*. *jamaicensis*, and *Sarcocystis* sp. Rod3 forming sarcocysts in rodents [18,30]. In the present work using the GsSstrF1/GsSstR1 primer pair, we amplified four *28S* rRNA genotypes of *S*. cf. *strixi* differing by 0.2–0.6% (Table 2). Given the findings outlined above, such differences may also be on an interspecific level. Nevertheless, to discriminate *Sarcocystis* species, it is necessary to test these parasites in IHs. Recently, an increasing number of unnamed *Sarcocystis* spp. presumably using rodents as their IHs have been identified. However, such *Sarcocystis* spp. are detected in DHs or in extra-intestinal tissues of IHs, but not from individually isolated sarcocysts [18,29,85,86,87,88,89]. In light of what was discussed, the standard isolation and detailed morphological and molecular examination of sarcocysts are needed for the disclosure of *Sarcocystis* diversity in rodents.

In the current study, more *Sarcocystis* spp. using birds as IHs and DHs were detected compared with parasite species employing small mammals as IHs (Table 2). However, there are limited data on *Sarcocystis* species that use rodents as their IHs [90]. More than 40 *Sarcocystis* spp. have been identified in the muscles or brains of rodents, but the true number of these parasites is thought to be much higher [12,87,90]. It should be noted that the most extensive knowledge is accumulated on the *Sarcocystis* species forming sarcocysts in the house mouse (*Mus musculus*) and brown rat (*Rattus norvegicus*). By contrast, there is still a lack of data on the prevalence and richness of *Sarcocystis* spp. infecting wild mice and voles [90].

### 4.5. Identification of Sarcocystis spp. in Intestines of Naturally Infected Raptors

Raptors inhabit most ecosystem types, occupy large home ranges, feed at the top of the food web, and are highly sensitive to chemical contamination [91]. Due to particularly good indicators of environmental health, raptors are an important study object. However, half of the global raptor population is declining [7], making it difficult to study them as potential DHs of *Sarcocystis* spp. It is assumed that most *Sarcocystis* species are not harmful to raptors as DHs of these parasites [12]. However, raptors can spread *Sarcocystis* species that are pathogenic for their IHs [13,14,15,23,25,30]. In the current work, in the small intestines of raptors, we detected *S*. *glareoli* forming cysts in the brains of common voles and likely several other rodent species [12,18,29,83,92] and *S*. *halieti* potentially causing encephalitis for birds [51]. However, there is still a lack of studies on the identification and pathogenicity of *Sarcocystis* spp. found in the faecal matter in the intestines of raptors [13,14,23,25,30,34,35]. Hence, further investigations are needed to disclose the impact of raptors transmitting pathogenic *Sarcocystis* spp. and on the health of raptors themselves.

Sequencing is the most reliable and comprehensive way to obtain information on different *Sarcocystis* spp. in IHs and DHs from PCR fragments [93]. In the present study, *Sarcocystis* species were identified using species-specific or genus-specific nested PCR and direct Sanger sequencing of amplified DNA fragments. However, this approach has some limitations for disclosing whole *Sarcocystis* species richness in intestinal samples, as numerous studies have proven that a single raptor can carry more than one *Sarcocystis* species in its intestine [14,15,18,25,30]. In the current work, at least two different *Sarcocystis* species were identified in 72.5% of the raptors examined. Furthermore, using genus-specific primers, we obtained some sequences with double or poly peaks, likely indicating *Sarcocystis* co-infection [14]. New species forming sarcocysts in the muscles of birds, rodents, or predatory mammals are becoming increasingly described [13,17,21,26,37,38,46,47,50,69], and the *Sarcocystis* species richness is expected to be much higher. Therefore, in this work, using a dozen species-specific primers and several genus-specific primers, theoretically, does not allow the identification of all *Sarcocystis* species from birds, rodents, or predatory mammals present in the samples tested. Therefore, cloning of PCR products or metabarcoding can be an alternative way to reveal *Sarcocystis* species richness [27,94,95,96]. In summary, further development of sensitive molecular methods for the identification of *Sarcocystis* species is expected.

## 5. Conclusions

Based on the nested PCR of *28S* rRNA and *ITS1* sequences, we identified nine previously identified *Sarcocystis* species, including pathogenic *S*. *glareolid* and *S*. *halieti*, as well as *S*. *arctica*, *S*. *columbae*, *S*. *cornixi*, *S*. *kutkienae*, *S*. cf. *strixi*, *S*. *turdusi*, and *Sarcocystis* sp. ex *Corvus corax*, in the intestine mucosa of raptors from Spain. Furthermore, three genetically new species, *Sarcocystis* sp. Rod6, *Sarcocystis* sp. Rod7, and *Sarcocystis* sp. 22AvEs1, were detected. Moreover, new DHs of *S*. *arctica*, *S*. *columbae*, *S*. *cornixi*, *S*. *glareoli*, *S*. *halieti*, *S*. *kutkienae*, and *S*. cf. *strixi* were identified by means of molecular methods. The distribution of certain *Sarcocystis* species established was related to the diet preferences of the raptors analysed.

Surprisingly, four or more *Sarcocysts* species were detected in eight different investigated raptor species. Furthermore, at least two different *Sarcocystis* species were identified in 72.5% of the raptors tested. Thus, the raptors analysed play a significant role in the distribution of various *Sarcocystis* species. However, further development of the molecular distinguishment of *Sarcocystis* species in the intestines or faeces of naturally infected raptors should be developed.

## Figures and Tables

**Figure 1 animals-15-00646-f001:**
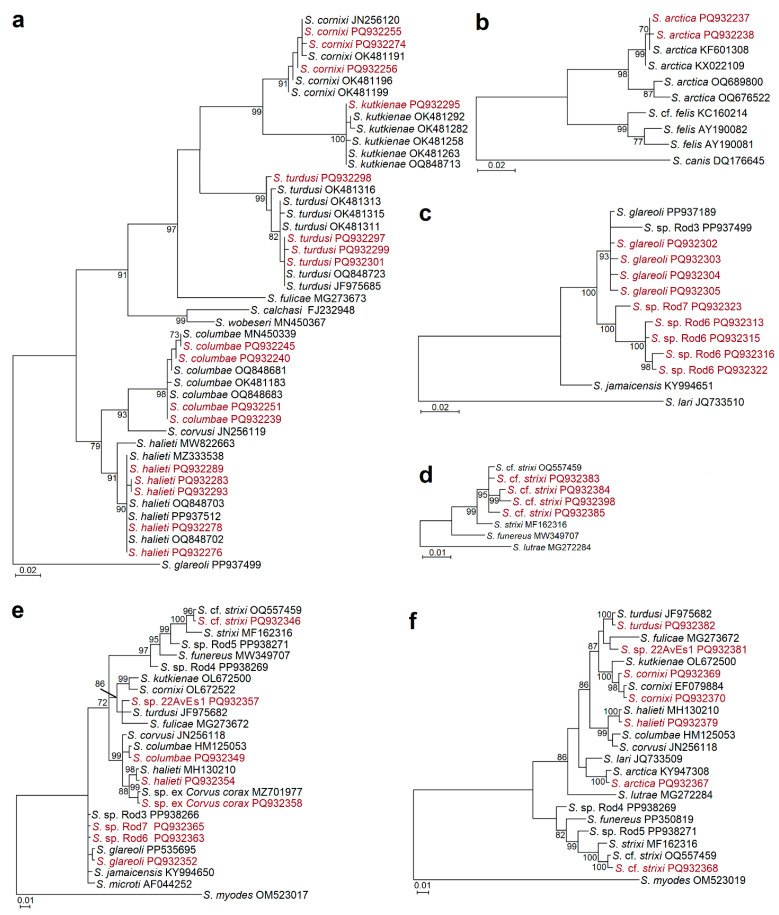
Phylogenetic relationships of selected *Sarcocystis* spp. based on *ITS1* (**a**–**c**) and *28S* rRNA (**d**–**f**). Phylogenetic trees were generated using Bayesian methods scaled according to the branch length and rooted on *S*. *glareoli* (**a**), *S*. *canis* (**b**), *S*. *lari* (**c**), *S*. *lutrae* (**d**), and *S*. *myodes* (**e**,**f**). Alignments contained 432 nucleotide positions and 46 sequences (**a**), 688 nucleotide positions and 10 sequences (**b**), 535 nucleotide positions and 13 sequences (**c**), 506 nucleotide positions and 8 sequences (**d**), 637 nucleotide positions and 26 sequences (**e**), and 626 nucleotide positions and 23 sequences (**f**). GTR+G (**a**), HKY+G (**b**,**d**–**f**), and HKY (**c**) models were used for analyses. Posterior probability values are shown next to branches. Our sequences are coloured dark red. Sequences were obtained using GsScolF/GsScolR, GsScornF2/GsScornR2, GsShalF/GsShalR2, and GsSkutkF2/GsSkutkR2, GsSturF/GsSturR (**a**), GsSarcF1/GsSarcR1 (**b**), GsSglajamF1/GsSglajamR1 (**c**), GsSstrF1/GsSstR1 (**d**), SgraupaukF/SgraupaukR (**e**), and Ssprod2F/Ssprod2R (**f**) primers. Only different genotypes of the same species identified in the present work are presented in phylograms.

**Figure 2 animals-15-00646-f002:**
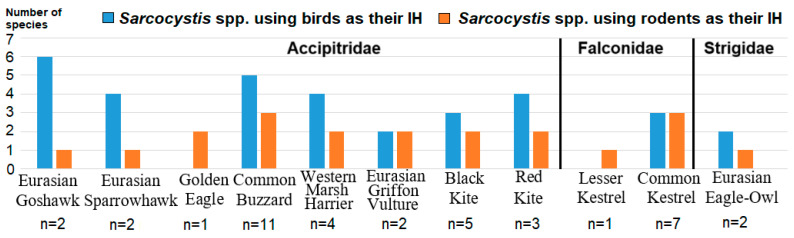
Number of *Sarcocystis* species identified in certain hosts. IH—intermediate host; DH—definitive host; n—number of birds examined.

**Table 1 animals-15-00646-t001:** List of oligonucleotides used for the identification of *Sarcocystis* spp.

Primer Name	Primer Sequence	T_a_	Step	Target Species	LP	Genetic Region
Sgrau281 [29]	Gaacagggaagagctcaaagtg	61	I	*Sarcocystis* spp.	898	*28S* rRNA
Sgrau282 [29]	Ggtttcccctgacttcattctac					
SgraupaukF [30]	Cgtatttgccctgtgtcctt	55	II	*Sarcocystis* spp.	659	*28S* rRNA
SgraupaukR [30]	Gtcgtaggtgcaaagcataacatc					
Ssprod2F [30]	Gtgcaaagcataacatcctttta	56	II	*Sarcocystis* spp.	645	*28S* rRNA
Ssprod2R [30]	Ggtgagagtcccgtatttgctc					
GsSstrF1 [PS]	Gaaatcgaagactcttgactgaatc	59	II	*S*. *strixi*	589	*28S* rRNA
GsSstrR1 [PS]	Aagagaaagagtgtagcccgatcat					
SU1F [31]	Gattgagtgttccggtgaattatt	57	I	*Sarcocystis* spp.	1100	*ITS1*
5.8SR2 [31]	Aaggtgccatttgcgttcagaa					
GsSarcF1 [23]	Caagcacaaatgtatcatcgtctta	61	II	*S*. *arctica*	524	*ITS1*
GsSarcR1 [23]	Tcctttttattctcaaatgacttcg					
GsSlutF1 [23]	Gaaacgtctgaaatgatgatggtat	59	II	*S*. *lutrae*	528	*ITS1*
GsSlutR1 [23]	Aagagaaaagaaaaacagccagac					
GsScalF2 [25]	Cttttgtaaggttggggacata	59	II	*S*. *calchasi*	583	*ITS1*
GsScalR2 [25]	Gcctccctccctctttttg					
GsScolF [23]	Atatgttcatcctttcgtagcgttg	65	II	*S*. *columbae*	579	*ITS1*
GsScolR [23]	Gccatccctttttctaagagaagtc					
GsScornF2 [23]	Agttgttgacgttcgtgaggtc	61	II	*S*. *cornixi*	483	*ITS1*
GsScornR2 [23]	Acacactactcattatctcctactcct					
GsScovF [23]	Tattcattctttcggtagtgttgag	59	II	*S*. *corvusi*	524	*ITS1*
GsScovR [23]	Ttactcttttaacagcttcgctgag					
GsSfulF [23]	Caaagatgaagaaggtatatacgtgaa	59	II	*S*. *fulicae*	449	*ITS1*
GsSfulR [23]	Ctttactcttgaagaacgacgttga					
GsShalF [23]	Gataattgactttacgcgccattac	65	II	*S*. *halieti*	644	*ITS1*
GsShalR2 [23]	Ccatccctttttctaaaggaggtc					
GsSkutkF2 [23]	Acacacggtcgagttgatatgac	61	II	*S*. *kutkienae*	625	*ITS1*
GsSkutkR2 [23]	Tctttacccttaaacaatttcgttg					
GsSlarF [23]	Ttcgtgaggttattatcattgtgct	59	II	*S*. *lari*	545	*ITS1*
GsSlarR [23]	Ggcgatagaaatcaaagcagtagta					
GsSturF [23]	Gatttttgatgtccgttgaagttat	61	II	*S*. *turdusi*	561	*ITS1*
GsSturR [23]	Cattcaaatatgctctcttccttct					
GsSwobF [23]	Atgaactgctttttcttccatcttt	58	II	*S*. *wobeseri*	532	*ITS1*
GsSwobR2 [23]	Ctcctcttgaaggtggtcgtgt					
GsSglajamF1 [30]	Tttcgtagcgctgaggagatt	57	II	*S*. *glareoli* *S*. *jamaicensis* *S*. sp. Rod 3	~560	*ITS1*
GsSglajamR1 [30]	Tgcttttcttcctttacttttgaatg					

T_a_—annealing temperature expressed in °C; LP—length of the product in bp.

**Table 2 animals-15-00646-t002:** Genetic identification of *Sarcocystis* spp. in intestinal scrapings from raptors.

Species	Length of Fragment	Number of Sequences	Genotypes, Differences Between Genotypes	Similarity with Certain Species	Similarity with the Most Closely Related Species
*ITS1* region, species-specific primers					
*S*. *arctica*	474 bp	2	2, 0.2%	96.4–100%	88.2–89.6% *S*. *felis*
*S. columbae*	529 bp	16	4, 0.2–0.8%	99.1–100%	95.9–96.4% *S*. sp. isolate 38P, 95.7–96.2% *S*. sp. ex *Accipiter cooperii*
*S. cornixi*	435 bp	21	3, 0.2–0.5%	98.9–100%	89.4–90.5% *S*. *kutkienae*
*S. halieti*	595-598 bp	19	5, 0.2–0.8%	98.3–100%	96.1–96.6% *S*. sp. isolate Skua-2016-CH
*S. kutkienae*	578 bp	2	1, N/A	99.3–100%	88.9% *S*. *cornixi*
*S. turdusi*	511-513	5	4, 0.2–1.9%	97.9–100%	85.1–85.9% *S*. *kutkienae*
*ITS1* region, GsSglajamF1/GsSglajamR1					
*S*. *glareoli*	518 bp	11	4, 0.2–0.4%	99.8–100%	98.3–98.5% *S*. sp. rod3
*S*. sp. Rod6	515 bp	10	4, 0.2–0.8%	N/A	95.8–96.5 *S*. *glareoli*
*S*. sp. Rod7	511 bp	2	1, N/A	N/A	96.2–96.4% *S*. *glareoli*
*28S* rRNA, GsSstrF1/GsSstR1					
*S*. cf. *strixi*	539 bp	20	4, 0.2–0.6%	99.6–100%	99.1–99.4% *S*. *strixi*, 97.4–98.0% *S*. *funereus*
*28S* rRNA, Ssprod2F/Ssprod2R					
*S. cornixi*	622 bp	10	2, 0.2%	99.8–100%	98.9–99.0% *S*. *kutkienae*
*S*. *turdusi*	623 bp	1	1, N/A	100%	98.7% *S*. *cornixi*
*S. arctica*	614 bp	1	1, N/A	99.4–100%	97.6% *S*. *lari*
*S. halieti*	619 bp	2	1, N/A	99.8–100%	99.2% *S*. *corvusi*, *S*. sp. ex *Corvus corax*
*S.* sp. 22AvEs1	622 bp	1	1, N/A	N/A	98.9% *S*. *turdusi*
*S*. cf. *strixi*	626 bp	1	1, N/A	99.8%	98.7% *S*. *strixi*
*28S* rRNA, SgraupaukF/SgraupaukR					
*S*. cf. *strixi*	623 bp	3	1, N/A	99.8%	98.7% *S*. *strixi*
*S*. *glareoli*	616 bp	2	1, N/A	99.8%	99.7% *S*. *jamaicensisi*, 99.4% *S*. sp. Rod3
*S*. sp. Rod6	616 bp	2	1, N/A	N/A	99.7% *S*. *jamaicensis*, 99.5% *S*. *glareoli*, *S*. sp. rod3
*S*. sp. Rod7	616 bp	2	1, N/A	N/A	99.8% *S*. *jamaicensis*, 99.7% *S*. *glareoli*, *S*. sp. Rod3
*S. halieti*	616 bp	3	1, N/A	99.8–100%	99.2% *S.* sp. ex *Corvus corax*
*S. columbae*	616 bp	3	1, N/A	100%	99.5% *S*. *corvusi*
*S.* sp. ex *Corvus corax*	616 bp	5	1, N/A	100%	99.4% *S*. *corvusi*
*S.* sp. 22AvEs1	616 bp	1	1, N/A	N/A	98.8% *S*. *turdusi*

N/A—not applicable.

**Table 3 animals-15-00646-t003:** The detection of *Sarcocystis* species in raptors examined.

Host Species	n	Prevalence of *Sarcocystis* spp., %	*Sarcocystis* Species Identified	
			$\overset{-}{x}$	List of Species
Eurasian Goshawk	2	100	5	*S*. *columbae*, *S*. *cornixi*, *S*. *glareoli*, *S*. *halieti*, *S*. *kutkienae*, *S*. *turdusi*, *S*. sp. ex *Corvus corax*
Eurasian Sparrowhawk	2	50	2.5	*S*. *cornixi*, *S*. *glareoli*, *S*. *halieti*, *S*. *turdusi*, *S*. sp. ex *Corvus corax*
Golden Eagle	1	100	2	*S*. sp. Rod6, *S*. cf. *strixi*
Common Buzzard	11	90.9	3.2	*S. arctica*, *S*. *columbae*, *S*. *cornixi*, *S*. *glareoli*, *S*. *halieti*, *S*. *turdusi*, *S*. cf. *strixi*, *S*. sp. ex *Corvus corax*, *S*. sp. Rod7
Western Marsh Harrier	4	100	4.3	*S*. *columbae*, *S*. *cornixi*, *S*. *halieti*, *S*. cf. *strixi*, *S*. sp. Rod6, *S*. sp. 22AvEs1
Eurasian Griffon Vulture	2	100	2.5	*S*. *columbae*, *S*. *glareoli*, *S*. *halieti*, *S*. cf. *strixi*
Black Kite	5	80.0	3.2	*S*. *columbae*, *S*. *cornixi*, *S*. *halieti*, *S*. cf. *strixi*, *S*. sp. Rod6
Red Kite	3	66.7	2	*S. arctica*, *S*. *columbae*, *S*. *cornixi*, *S*. *halieti*, *S*. *kutkienae*, *S*. cf. *strixi*, *S*. sp. Rod6
Lesser Kestrel	1	100	1	*S*. cf. *strixi*
Common Kestrel	7	57.1	1.9	*S*. *columbae*, *S*. *cornixi*, *S*. *glareoli*, *S*. *halieti*, *S*. cf. *strixi*, *S*. sp. Rod6
Eurasian Eagle-Owl	2	100	2	*S*. *columbae*, *S*. *cornixi*, *S*. cf. *strixi*

n—number of birds examined; $\overset{-}{x}$—average number of *Sarcocystis* species detected per sample.

## Data Availability

Data supporting the conclusions of this article are included in the article. The sequences generated in the present study were submitted to the GenBank database under accession numbers PQ932237–PQ932324 and PQ932346–PQ932402.

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
