# Peer review of "Molecular Confirmation of Raptors from Spain as Definitive Hosts of Numerous Sarcocystis Species"

_animals, 2025, doi:10.3390/ani15050646_

Round 1
Reviewer 1 Report
Comments and Suggestions for Authors
This paper thoroughly examines the species of Sarcocystis spp. from the intestines of raptor birds in Spain using molecular methods. In the experiment, molecular methods were used to detect general genes, as well as species-specific genes, with a large number of different primers, which significantly increased the accuracy of the identified species. The phylogenetic analysis was done in detail and thoroughly. The discussion is very clearly presented in logical sections.
The attached PDF document contains minor changes to be made and a few comments for the authors.
General comment: The discussion should include info that refers to potential pseudoparasitism, considering the diet of the raptor species in which they were found (especially parasitic species that were identified for the first time in those birds). In general, it should be clarified how, based only on the DNA found in intestinal scrapings, we can claim that it originates from oocysts/sporocysts, and that the parasite did not accidentally enter the bird's digestive tract due to its diet.
In general, the paper should be published after the changes have been made.

Author Response
Point 1. This paper thoroughly examines the species of Sarcocystis spp. from the intestines of raptor birds in Spain using molecular methods. In the experiment, molecular methods were used to detect general genes, as well as species-specific genes, with a large number of different primers, which significantly increased the accuracy of the identified species. The phylogenetic analysis was done in detail and thoroughly. The discussion is very clearly presented in logical sections.
Response 1. Thank you very much for such a positive evaluation.
The attached PDF document contains minor changes to be made and a few comments for the authors.
Point 2. Supernatant or sediment? Since the supernatant was discarded in the first step, it is not clear why it is used as a medium for investigation, after being discarded 3 times. I suppose its a typing error.
Response 2. Thank you for the error you have noticed. Yes, it is a typing error, must be a “precipitate”. We changed it.
Point 3. Please, write the full name of the substance in brackets
Response 3. Thank you for your valuable comment. We added the missing information: “(HBSS Hanks' Balanced Salt Solution)”.
Point 4. I think this part should be explained a bit more, for it to be clear for all the readers. Since molecular analysis was the aim of the study it wasn't necessary to do microscopic examination, but if it was done, the limitations should be discussed.
Response 4. Thank you for your meaningful comment. Prof. Luis Monteagudo (Department of Anatomy, Embryology and Animal Genetics, Veterinary Faculty, Zaragoza, Spain) allowed us to prepare the intestinal samples of raptors in him laboratory (we are very grateful to him), but unfortunately, the laboratory lacked some of the necessary facilities, so it was not possible to implement all of the recommendations proposed by Verma et al. Once we have prepared the intestines with the modifications that we have not used before, we need to check that the method still works and that we see the sporulated oocysts/sporocysts. Due to technical difficulties and lack of time, only three bird intestinal samples were microscopically examined for Sarcocystis spp. However, we realise that this is becoming confusing to understand in the text and, based on comments from other reviewers, we have decided to remove the microscopic part from our paper. We deleted all microscopic sentences from Materials and Methods, Results and Discussion.
Point 5. So, were these 3 samples included in the study or not?
Response 5. Thank you for your comment. These three samples were included in the study. But as I mentioned previously, we realise that this is becoming confusing to understand in the text and, based on comments from other reviewers, we have decided to remove the microscopic part from our paper. Your remark that the purpose of this study is molecular analysis and not microscopic examination is very appropriate. Thank you one more time.
Point 6. Please, include method used for detection, and magnification.
Response 6. Thank you for your meaningful comment. We have deleted Figure 1.
Point 7. The authors should consider discussing possible pseudoparasitism for some species, based on previous investigations and data regarding definitive and intermediate hosts for some species, bcs detection of DNA in intestinal scrapings not necessarily needs to mean that the oocysts were formed in the examined animal.
Response 7. Thank you for your valuable comment. Your point about pseudoparasitism is very appropriate, so we have added sentences in lines 283-289: “However, the possible determination of pseudoparasitism for some Sarcocystis species found in intestinal scrapings of raptors cannot be excluded. Based on previous inves-tigations and data regarding DHs and IHs for some species, the detection of Sarcocystis DNA in intestinal scrapings does not necessarily mean that the oocysts/sporocysts were formed in the examined birds. The recovered DNA may also belong to Sarcocystis spp. that were in the bird's feeding carcass [13,23]”.
Point 8. Why was this done? Were these samples included in the molecular analysis or not? If not, why?
Response 8. Thank you. These 3 samples were included in the study. We realise that this is becoming confusing to understand in the text and, based on comments from other reviewers, we have decided to remove the microscopic part from our paper.
Point 9. Line 307. Sarcocystis
Response 9. Thank you. We changed.
Point 10. General comment: The discussion should include info that refers to potential pseudoparasitism, considering the diet of the raptor species in which they were found (especially parasitic species that were identified for the first time in those birds). In general, it should be clarified how, based only on the DNA found in intestinal scrapings, we can claim that it originates from oocysts/sporocysts, and that the parasite did not accidentally enter the bird's digestive tract due to its diet.
In general, the paper should be published after the changes have been made.
Response 10. Thank you for your meaningful comment. We completely agree with you and have added sentences in lines 283-290: “However, the possible determination of pseudoparasitism for some Sarcocystis species found in intestinal scrapings of raptors cannot be excluded. Based on previous inves-tigations and data regarding DHs and IHs for some species, the detection of Sarcocystis DNA in intestinal scrapings does not necessarily mean that the oocysts/sporocysts were formed in the examined birds. The recovered DNA may also belong to Sarcocystis spp. that were in the bird's feeding carcass [13,23]. Further transmission experiments are necessary to confirm whether the identified Sarcocystis species are transmitted by these raptorial birds“.
Reviewer 2 Report
Comments and Suggestions for Authors
The article presents the results of molecular analyses of the occurrence of protozoa of the Sarcocystis genus in definitive hosts, in this case birds of prey. In order to further emphasize the role of predators in the development cycle of Sarcocystis spp., I suggest making a slightly more detailed analysis of the invasiveness of this parasite. Considering that oocysts excreted by definitive hosts contaminate the environment - intermediate hosts are primarily herbivorous and omnivorous animals. On the other hand, the location of cysts (containing bradyzoites) concerns the tissues of intermediate hosts, the definitive hosts are carnivorous and omnivorous animals. Hence, birds of prey have an exceptionally high potential for infection. The work is very interesting in the context of the genetic diversity of individual Sarcocystis species. Due to the lack of comparable results of microscopic examinations in the research group, due to unknown problems, I suggest removing this part from the text and focusing only on the results of molecular analyses. Fully reliable microscopic examinations should be performed on samples of fresh stool or fresh intestinal content. This is especially true for the genus Sarcocystis, where oocysts are invasive (sporulated) already at the moment of excretion with the stool and easily disintegrate into sporoblasts (sporocysts). This is their differentiating feature from other protozoa from the subphylum Apicomplexa. The work is interesting, I suggest publishing it after minor corrections.
Author Response
Point 1. The article presents the results of molecular analyses of the occurrence of protozoa of the Sarcocystis genus in definitive hosts, in this case birds of prey. In order to further emphasize the role of predators in the development cycle of Sarcocystis spp. I suggest making a slightly more detailed analysis of the invasiveness of this parasite. Considering that oocysts excreted by definitive hosts contaminate the environment - intermediate hosts are primarily herbivorous and omnivorous animals. On the other hand, the location of cysts (containing bradyzoites) concerns the tissues of intermediate hosts, the definitive hosts are carnivorous and omnivorous animals. Hence, birds of prey have an exceptionally high potential for infection. The work is very interesting in the context of the genetic diversity of individual Sarcocystis species. Due to the lack of comparable results of microscopic examinations in the research group, due to unknown problems, I suggest removing this part from the text and focusing only on the results of molecular analyses. Fully reliable microscopic examinations should be performed on samples of fresh stool or fresh intestinal content. This is especially true for the genus Sarcocystis, where oocysts are invasive (sporulated) already at the moment of excretion with the stool and easily disintegrate into sporoblasts (sporocysts). This is their differentiating feature from other protozoa from the subphylum Apicomplexa. The work is interesting, I suggest publishing it after minor corrections.
Response 1. Thank you very much for such a positive evaluation. We completely agree with you and have removed microscopic part from our paper. We deleted all microscopic sentences, as well as Figure 1, from Materials and Methods, Results and Discussion. Also, we add some sentences about pseudoparasitism in lines 283-290: “However, the possible determination of pseudoparasitism for some Sarcocystis species found in intestinal scrapings of raptors cannot be excluded. Based on previous inves-tigations and data regarding DHs and IHs for some species, the detection of Sarcocystis DNA in intestinal scrapings does not necessarily mean that the oocysts/sporocysts were formed in the examined birds. The recovered DNA may also belong to Sarcocystis spp. that were in the bird's feeding carcass [13,23]. Further transmission experiments are necessary to confirm whether the identified Sarcocystis species are transmitted by these raptorial birds“.